# Teoloyucan Geomagnetic Observatory operation over a Quality Management System ISO 9001:2015

Ana Caccavari-Garza[1], Gerardo Cifuentes-Nava[1], Armando Carrillo-Vargas[1], Adriana Elizabeth Gonzalez-Cabrera[2], Charbeth López-Urías[1], and Juan Esteban Hernandez-Quintero ⚕ .

[1]Magnetic Service, Universidad Nacional Autónoma de México, Mexico City, 04510, Mexico
[2]Mexican Solarimetric Service, Universidad Nacional Autónoma de México, Mexico City, 04510, Mexico
⚕    Deceased

*Corresponding author*: Ana Caccavari-Garza (anacg@igeofisica.unam.mx)

**Abstract**. Geomagnetic observatories are essential for the study of the Earth's magnetic phenomena; they allow the precise and continuous measurement of the geomagnetic field. They are built and operated according to rigorous international standards to ensure the acquisition of high-quality geomagnetic data. Given the nature of Quality Management Systems (QMS) based on ISO 9001:2015, we consider that their implementation in a geomagnetic observatory can be a valuable tool that allows monitoring the follow-up of international standards and ensuring their proper operation, thus guaranteeing high-quality geomagnetic data.

Some of the main advantages of implementing a quality management system and obtaining an ISO 9001:2015 certification include setting clear objectives, systematically analyzing risks that could affect both functionality and data quality, fostering a culture of continuous improvement, promoting context analysis through a *strengths*, *weaknesses*, *opportunities,* and *threats* analysis, and strategic planning based on this knowledge. In addition, involving the senior management of the responsible institution can help raise awareness of the operation's characteristics and needs. It also facilitates the continuous monitoring of users' requirements and satisfaction, as well as the correct documentation of all procedures carried out for its operation.

This study presents the registered experience in the implementation of a QMS in the only magnetic observatory in Mexico: Teoloyucan. It outlines the operation of the observatory, including data acquisition platforms, transmission, reduction, management, and data publication. It also describes the process for implementing the quality management system in the data deployment procedures, highlighting its advantages, disadvantages, and challenges during its adoption.

Keywords: Geomagnetic observatories, quality management systems, quality data, improvement, geomagnetism

## 1.    Introduction

The objective of geomagnetic observatories is to record, continuously and in the long term, the temporal variations of the magnetic field vector, maintaining the absolute standard of accuracy of the measurements (Wienert, 1970; Jankowski and

Sucksdorff, Rasson, 2004; Bracke, 2025). The components of the magnetic field vector include total intensity (F), horizontal (H), vertical (Z), north (X), east (Y), declination (D), and inclination (I).

INTERMAGNET (International Real-time Magnetic Observatory Network) is a worldwide network of geomagnetic observatories managed by the IAGA (International Association of Geomagnetism and Aeronomy). It establishes guidelines and verifies that the observatories belonging to this network generate high-quality data. These observatories are important

because they provide data for both scientific research and practical applications (Rasson et al., 2011).

In Mexico, the only existing geomagnetic observatory is managed by the Magnetic Service of the Geophysics Institute of the National Autonomous University of Mexico (UNAM). It has operated in the town of Teoloyucan, State of Mexico (Figure 1), since August 1914, having previous operating locations in Mexico City from 1879 to 1912, but it had to be moved due to electric and magnetic noise introduced to those buildings. The Teoloyucan site was selected because it was remote enough

from the city and would not be affected by magnetic noise. It began its operation with Mascart variometers (D, H, Z) that belonged to the previous site. In 1923, the magnetometer Dover 123 (F) and the inclination compasses Fauth 73, Negretti-Zambra 65, and Chasselon 64 (I) were incorporated. Later, in 1931, the set of Eschenhagen photographic recording variometers (D, H, Z) from the Askania House was acquired and worked until the shift to the digital era (Hernandez Quintero et al., 2018). The observatory's magnetic values were originally published in 1950 (Sandoval, 1950).

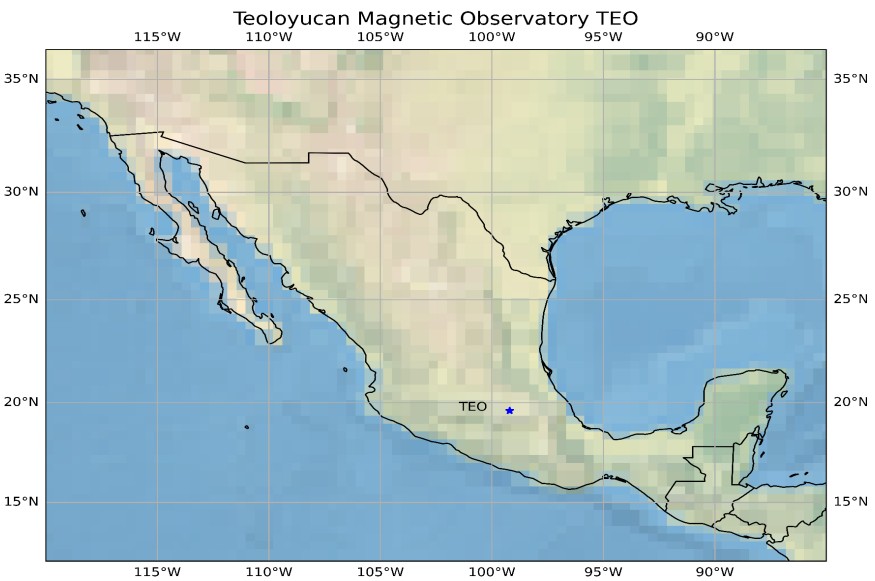


**Figure 1. Observatory location map (made with Cartopy and Natural Earth. Free vector and raster map data @ naturalearthdata.com)**

In 1978, it had to be relocated, moving 700 m to the southwest, to the position where it is currently located (99° 11' 35.735" W, 19° 44' 45.100" N, 2280 masl (Figure 1)). The present operational buildings were constructed during this period (Figure 2).

In 1996, it entered the digital era during the first Latin American Congress of Geomagnetic Instrumentation, which was hosted at the Magnetic Observatory of Teoloyucan, using a three-component fluxgate variometer (Declination (D), Inclination (I), Total Intensity (F), baptized as LAMA, from the Royal Meteorological Institute of Belgium, RMI) and a three-component fluxgate variometer FGE (DHZ, from the Danish Meteorological Institute DMI), along with a PPM Geometrics G856 magnetometer and a RUSKA non-magnetic theodolite converted to a DI-flux magnetometer for absolute observations (Figure 2). The Internet became accessible for data transmission in 2000 (Hernandez-Quintero et al., 2018).

Since this date, it continued operating without major changes until 2021, when the variometers' installation was affected by an electrical failure, and they were replaced by a LEMI 025 variometer. In addition to this change and following the new *quality management* guidelines of the National Autonomous University of Mexico, it was decided to incorporate the "geomagnetic data deployment in near real time" procedure into the Quality Management System (QMS) of the geophysical services of the Institute of Geophysics of UNAM.

The observatory has a continuous record from 1914 until now; it was part of the INTERMAGNET observatories from 2002 to 2008, but due to instrumental issues, it was not able to maintain data quality, even if it continued operating and sending data to INTERMAGNET until 2021. In 2022 it began operating TEO currently operates with a Ukrainian LEMI025 fluxgate (XYZ) variometer (by Lviv Center of Space Research Institute), a GSM90 Total Intensity (F) magnetometer (by GEM Systems), and a Zeiss Theo 020B DI-flux (Declination (D), Inclination (I)) theodolite (ZEISS/RMI), following INTERMAGNET standards (Wienert, 1970; Jankowski and Sucksdorff 1996; Rasson, 2004; Bracke, 2025), and relying on a quality management system to verify the implementation of these standards, with the intention of applying to INTERMAGNET.

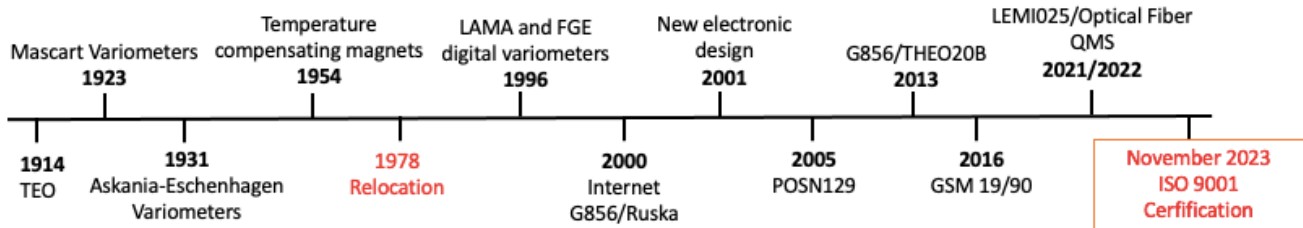

Figure 2. TEO's timeline. Location and instrumental changes. Scalar magnetometers: G856 (Geometrics), GSM19/90 (GemSystems), POSN129. Variometers: Mascart, Askania-Eschenhagen, LAMA (RMI), FGE (DMI). DI-flux: Ruska, Zeiss Theo 020B (RMI).

It is important to emphasize that implementing a QMS does not define the operating guidelines themselves; these are given by the international standards already mentioned. The purpose of a QMS is to verify that these guidelines, which are described throughout the QMS, are rigorously followed.

This paper will present the aspects that we consider relevant when implementing a quality management system focused on the operation of a magnetic observatory: what a QMS is, how to adapt the work with a management system (define scope, objective, and quality policy), staff participation (experience, adaptation, profiles, and distribution of responsibilities), continuous review of *strengths, weaknesses, opportunities,* and *threats (SWOT)*, risk analysis, and maintenance scheduling. Additionally, the participation in the QMS of both top management and users and the advantage of having annual audits to 80 have a revision of the entire system. Lastly, we list a few things that we believe are disadvantages or arguments against a QMS.

**Methodology**

From the analysis of the ISO 9001:2015 Standard of the International Organization for Standardization, a quality management system is a set of principles (norms, procedures, and standards) required to develop the essential activity of an organization or work group under specific quality policies and objectives (Godinez-Mendez, 2023; International Organization for 85 Standardization, 2015). The documentation of the QMS could be summarized in the pyramid shown in Figure 3. Level 1-Quality manual: The mission, goals, objectives, and policy statements. Level 2-Quality system procedure: that describes quality control, validation, and process improvement. Level 3-Instructions (Standard Operation Procedures): operational work instructions and Level 4-Forms: Records, reports, and all the supporting records and forms associated with the operational procedure (Berte and Nevalainen, 1996).

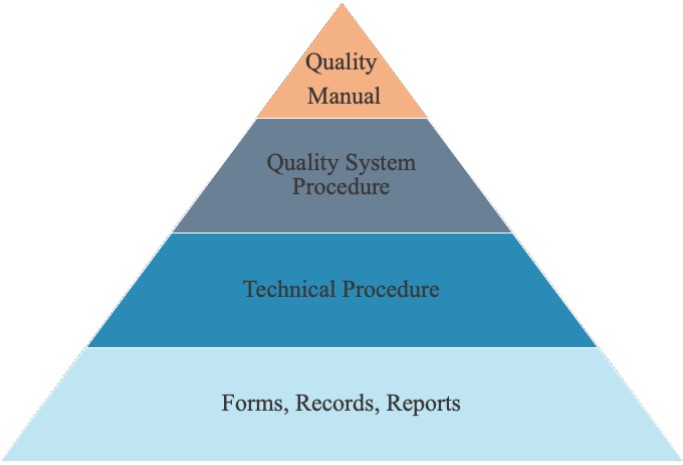

**Figure 3. Structure of a Quality Management System (adapted from Berte and Nevalainen, 1996)**

## 2.1 Work Adaptation with a Quality Management System

Performing a work task within a QMS does not change the work itself, but the vision one has about it. One becomes aware of the importance of each step of the process (Godinez-Mendez, 2023). As previously stated, the observatory has been in operation since 1914 in accordance with the established (Wienert, 1970; Jankowski and Sucksdorff, 1996; Rasson, 2004; Bracke, 2025). While implementing a QMS does not alter how it operates, it does enable the integration of continuous analysis of its operations, systematization of actions carried out, including preventative maintenance, improved role definition for each member, and the pursuit of continuous improvement, among other things.

The *quality manual* is a fundamental document that consolidates and communicates the structure, scope, and essential elements of the QMS. Although its development is no longer a mandatory requirement under ISO 9001:2015, it remains a valuable tool to facilitate internal understanding of the system, support staff training, and serve as a reference point for audits and external stakeholders. Its usefulness lies in its ability to integrate key processes, reflect the organization's commitment to quality, and contribute to the standardization and continual improvement of activities, thus strengthening the effectiveness of the management system.

## 2.2 Organization and top management context analysis: defining policies, objectives, and system scope.

The QMS *quality manual* describes the organization's context, in which the observatory is situated. The Magnetic Observatory of Teoloyucan is managed by the Magnetic Service of the Institute of Geophysics of the National Autonomous University of Mexico (UNAM), which is one of the six Geophysical Services of the Institute of Geophysics. The director of the Geophysics Institute and the director's representative constitute the senior management positions.

The Quality Management System was implemented for five of the six Geophysical Services, which collaboratively created the following *quality policy*:

"According to the International Standard ISO 9001:2015 quality management systems, geophysical services are committed to ensuring quality in the services they provide by offering truthful and reliable results to meet the needs and expectations of our customers and users with the ongoing commitment of the management to support the continuous improvement of processes as well as the effectiveness of the QMS with a risk-based approach."

The importance of proper observatory operation for obtaining and deploying high-quality geomagnetic data was the basis for the Magnetic Service's definition of the QMS's scope. Data reduction, transmission, deployment, and acquisition (infrastructure, instrumentation, and operation) are all included in this scope.

*Scope:* Geomagnetic data recorded at the Magnetic Observatory of Teoloyucan, State of Mexico, are published on the website.
In accordance with ISO 9001 requirements.

*Quality Objective* To publish on the website, geomagnetic data recorded at the Magnetic Observatory of Teoloyucan, State of Mexico, with at least 90% completeness.

As mentioned above, QMS systems encourage senior management to become involved in and committed to the system. The quality manual outlines its commitments to the annual review of the QMS, which includes updating the QMS's objectives,
processes, equipment, and anything else required to ensure accountability in terms of its effectiveness; guaranteeing that the QMS's quality policies and objectives are aligned with the Institute's context and strategic direction; and ensuring that any required assets are available.

Roles, responsibilities, authority, risk management, and SWOT are also covered in the *quality manual*; however, because these subjects are particularly important, they will be discussed in further detail below.

**2.3 SWOT and risk analysis**

Beyond the QMS, *SWOT* analysis is known to be a powerful tool for the analysis of an organization (Otero and Gache, 2006). The SWOT analysis is a combination of an assessment of an organization's external evaluation (i.e., opportunities and threats) and internal situation (i.e., strengths and weaknesses). It is also a tool that provides a summary of the strategic position of a given organization (Novy and Wahab, 2020).

The *SWOT* analysis carried out for the Magnetic Service:

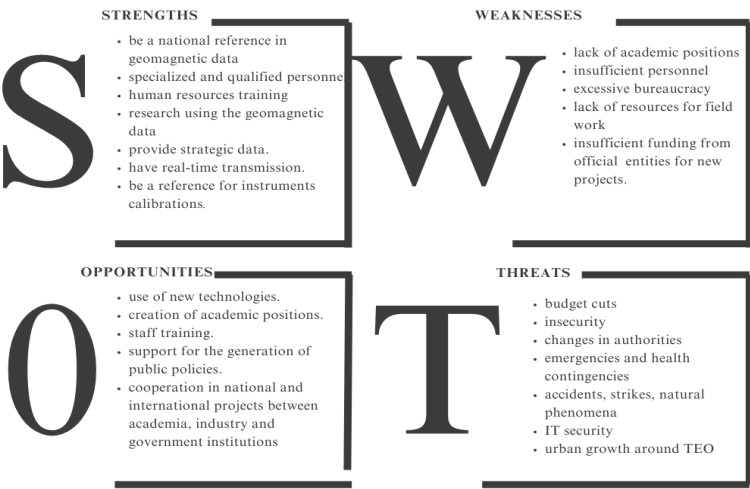

**STRENGTHS**
- be a national reference in geomagnetic data
- specialized and qualified personnel
- human resources training
- research using the geomagnetic data
- provide strategic data.
- have real-time transmission.
- be a reference for instruments calibrations.

**WEAKNESSES**
- lack of academic positions
- insufficient personnel
- excessive bureaucracy
- lack of resources for field work
- insufficient funding from official entities for new projects.

**OPPORTUNITIES**
- use of new technologies.
- creation of academic positions.
- staff training.
- support for the generation of public policies.
- cooperation in national and international projects between academia, industry and government institutions

**THREATS**
- budget cuts
- insecurity
- changes in authorities
- emergencies and health contingencies
- accidents, strikes, natural phenomena
- IT security
- urban growth around TEO

**Figure 4. SWOT:** *strengths, opportunities, weaknesses,* **and** *threats* **analysis for the Magnetic Service**

As shown in Figure 4, this analysis offers different possibilities, including enabling us to know our strengths and working to expand them through the opportunities. However, one of its most significant contributions is that it helps us identify the risks that our system may face by analyzing its weaknesses and threats. Based on this, we can then develop mitigation actions for these risks. An example of this in the context of TEO is the threat of lack of replacement instruments; with the weakness of long times in the purchasing processes, which might have a significant impact on observatory operations. From these analyses, the risk matrix is constructed, which includes the risks, the probability of their occurrence, the severity in case of occurrence, and the mitigation actions. This matrix is analyzed periodically, and it is observed whether the proposed risks occurred and whether the actions were relevant to mitigate them, and if not, new actions are proposed. Figure 5 shows a reduced version of the *risk matrix* obtained for TEO.

| Risk | Event that can cause | Probability/ frequency | Consequence /damage | Risk assessment (F vs D) | Mitigation actions | Occured | Consequence | Risk reassessment (F vs D) | New mitigation actions |
|---|---|---|---|---|---|---|---|---|---|
| Pandemic/strikes | No access to the facilities | Probable | Critical | 9 | Alternate center for downloading and processing information | No | None | 6 | None |
| Lack of supplier inventory | Not being able to replace faulty equipment/Stop recording data | Probable | Critical | 9 | Look for another provider | No | None | 6 | None |
| Long acquisition times | Not being able to replace faulty equipment/Stop recording data | Probable | Critical | 9 | Constant communication with the administration | No | None | 6 | None |
| Extreme phenomena | Stop recording data | Remote | Catastrophic | 8 | Backup equipment | No | None | 8 | None |
| Theft/vandalism | Stop recording data | Remote | Catastrophic | 8 | Surveillance and access control | No | None | 8 | None |
| Magnetic noise | Noisy data | Remote | Critical | 6 | Constant data quality control and surveys to detect Magnetic objects | No | None | 6 | None |
| Instrumental damage | Stop recording data | Probable | Catastrophic | 12 | Backup instruments | No | None | 8 | None |
| Lack of internet for streaming | Real-time reporting of data cannot be performed | Probable | Critical | 9 | Have a second way to download data | Yes | It affected the website display time | 9 | Have a second way to download data and request support from RedUNAM |
| Logical errors on the server | Real-time reporting of data cannot be performed | Probable | Critical | 9 | Second server that can perform the processing | No | None | 9 | None |
| Power outage | Real-time reporting of data cannot be performed | Probable | Critical | 9 | Backup power system in case of prolonged power outages | No | None | 9 | None |

**Figure 5. Reduced risk matrix of the Magnetic Service**

**2.4 The knowledge, experience, and skills of the personnel that will be integrated into the system.**

It is of utmost importance to consider the experience and knowledge of the personnel working in the magnetic observatory. All this knowledge must be documented. One of the main advantages of implementing a QMS is to document the procedures performed to preserve all the knowledge and experience of the personnel involved. Often, this knowledge belongs to a single person, who, when absent or retired, takes it with him/her, making it difficult to continue with the operation of the observatory. Later, the document where the whole operation is depicted will be described: the *technical procedure* for the deployment of

TEO geomagnetic data, where the responsibilities of the members of the QMS are described, and the job profiles specify the functions and requirements of each member of the group. Defining this contributes to better organization, performance, and commitment to the tasks. For the operation of the Magnetic Observatory of Teoloyucan, the Magnetic Service defined the following organization chart (Figure 6):

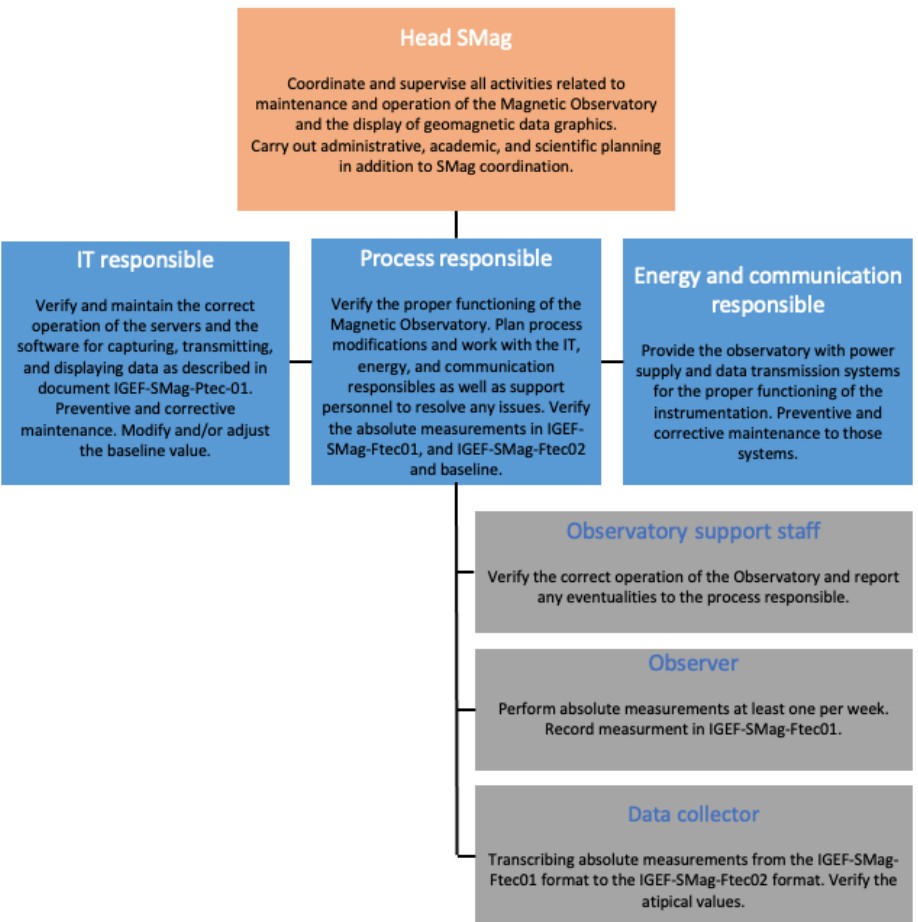

**Figure 6. Magnetic Service organizational chart**

## 2.5 Instrumentation and maintenance. Records

Within the *management documents*, in the part corresponding to *instrumental control,* there is a format (SMag IGEF-FPG02-01) that contains the inventory of instruments and servers, their characteristics, and their location. There is also a format (SMag IGEF-FPG02-03) to track the instrument maintenance, such as routine cleaning of servers, antivirus revision, delta F measurement, theodolite maintenance, comparison with Theo 010A DI-flux, cleaning, verification and adjustment, and internal cleaning of LEMI 025 memory, among others (Bracke, 2025). The instruments maintenance format includes the equipment, activity, responsible, date of last and next maintenance; it is also linked to a calendar to remind the activity.

Complementary to the records of the equipment and its maintenance, there is a logbook to record failures that occur in the equipment or systems, record any maintenance activity, either preventive or corrective, and record changes or events that produce disturbances in data that affect data quality.

Keeping this control allows us to carry out another key point in the QMS: the *planning of changes*, *corrective actions,* and *improvements*.

## 2.6 Corrective and preventive actions: keys to continuous improvement

Continuous improvement is one of the significant QMS approaches; organizations should focus on optimizing their operations, eliminating nonconformities, and preventing difficulties before they occur. *Planning changes*, *corrective actions*, and *improvements* are crucial processes for any organization's expansion and effectiveness.

*Corrective actions* are focused on fixing existing problems, addressing the root cause to prevent recurrence. The implementation stages consist of identifying the nonconformity or problem, analyzing the root cause, making a corrective action plan, implementing this plan, and verifying its effectiveness (Summer, 2025). An example of a corrective action in the Magnetic Service (AC-SMag-2024-05). The action was implemented, and its effectiveness was verified.

*Improvements*, on the other hand, seek to optimize processes, products, or services, generating added value for the organization. This can be illustrated in the Observatory's QMS, as it was considered practical to include knowledgeable staff in the design of energy and transmission systems within the first year of deployment. Planning for this change was therefore completed, the improvement was documented (MM-SMag-2023-02), and it was thereafter implemented and verified since it helps the Observatory run more efficiently, so it was a positive improvement.

In both, corrective actions and improvements, once the problem or possible improvement has been identified, the changes are carried out in a planned manner, outlining the following phases of change planning: Examine potential effects, available resources, and possible risks; establish the goal; and develop an organized action plan that outlines activities and

responsibilities, deadlines and assesses the change's efficacy. All this information is recorded and documented in the appropriate formats.

### 2.7 Stakeholders. Users and focus groups

Monitoring users and stakeholders is another benefit to consider when having a QMS. Stakeholders are not only individuals; they may also be groups and organizations that are interested in or impacted by the organization's QMS-related activities. Suppliers, supervisors, and administrative systems, in addition to the users themselves, may serve as examples of this. Each of 195 these stakeholders' relevance will be evaluated by the QMS, which will also monitor whether stakeholder requirements are being satisfied over time.

In the case of users, the QMS seeks to follow up on the satisfaction of their needs, to know their requirements, and to know their opinion about the products or services offered by the QMS. Users are those who visit the page or utilize the data produced by the magnetic observatory and its geomagnetic data deployment process. Every year, meetings with various users include a 200 focus group to review the system's efficacy, the availability of data, and issues deemed relevant for system improvement. A customer satisfaction survey can also be used to do this assessment.

### 2.8 Audits

Besides being a standard requirement, audits enable confirming if the QMS complies with both its own and the standard's requirements, as well as whether its deployment and operation are successful. Even though it could seem like a laborious 205 process at first, having these systematic evaluations allows one to ensure that every item outlined in the technical procedure and the QMS is promptly addressed (Kaziliūnas, 2010), which helps ensure proper observatory operation. To receive the ISO 9001:2015 certification, the Magnetic Service QMS undergoes two yearly audits: an external audit conducted by national certifying firms with international validity and an internal audit conducted by a university agency. There have been two external and three internal audits of our QMS. In November 2023, certification was acquired, and is renewed annually.

### 2.9 Disadvantages

Although this article covers most of the required documents, there are some whose purpose may be deemed less essential but still need to be completed. This is unquestionably one of the arguments against implementing a QMS. The staff may need to invest some time in this. It is crucial to note, that after the entire system is integrated, the documents that need to be updated or made are connected to the technical procedure and contribute, in this case, to the correct functioning of the observatory.

It is also important to mention that certification comes with a price, which in this instance was covered by management, weighing the advantages of using a QMS and certification in operations. Implementing a QMS without acquiring certification is one method to address this if you do not want to allocate this resource.

## 3    Results

### 3.1 Technical Procedure

The most relevant document of the QMS is the *technical procedure*, which details the observatory operation, beginning with the most crucial concepts' definitions, personnel responsibilities, infrastructure, instruments, and power and communication equipment, as well as the description of the activities, including the absolute measurements formats and data reduction. An overview of the key elements of the observatory's *technical procedure* "Graphical display of geomagnetic data", is provided below.

A schematic drawing of the Teoloyucan Magnetic Observatory is shown in Figure 7: absolute and variometer houses, two auxiliary huts, and an office.

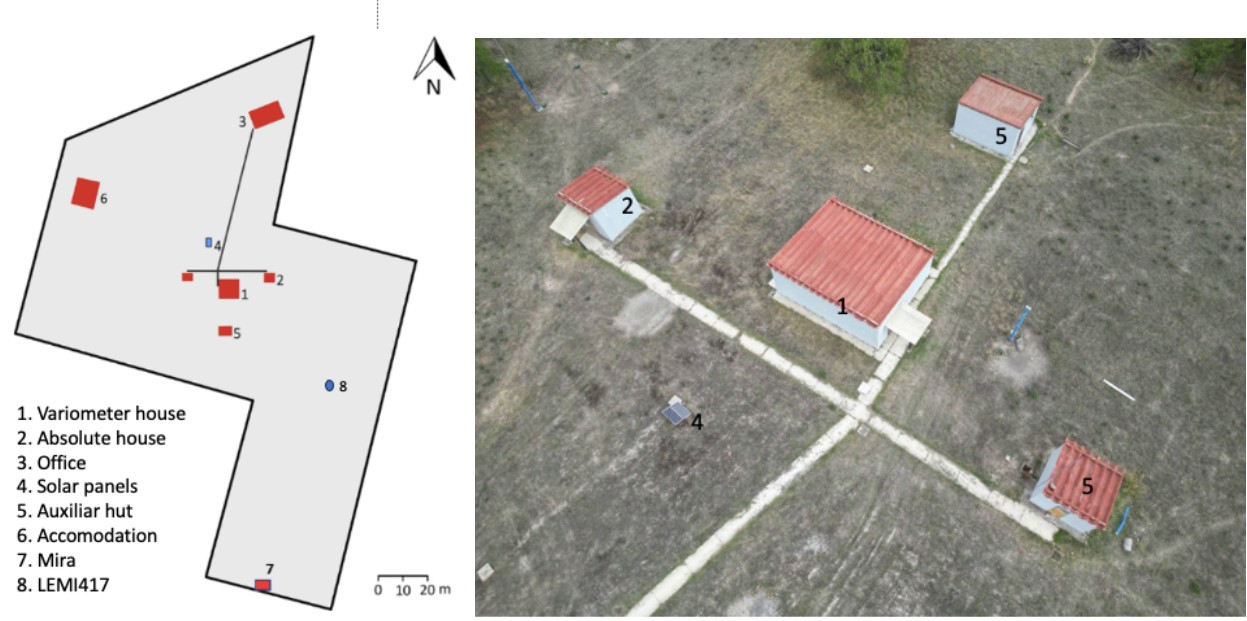

**Figure 7. a) TEO's schematic drawing and b) aerial photo**

    The observatory was built following the established standards: the absolute observations house has two pillars for
measurements, is built of non-magnetic materials, and has a window to observe the azimuth mark, located 70 m away; the

variometer house consists of an outer corridor and an inner isolated room. The inner room is equipped with five solid pillars and is built of non-magnetic materials. In one of those pillars is the LEMI 025 sensor. In the outer corridor is located the electronics of the LEMI 025, in the northern part the GSM90 magnetometer, and on the other side the power and transmission systems. The instruments are powered by solar panels (Figure 7). The data are transmitted by optic fiber to the office since 2022 to avoid problems due to currents induced by lightning strikes. Instruments are shown in Figure 8.


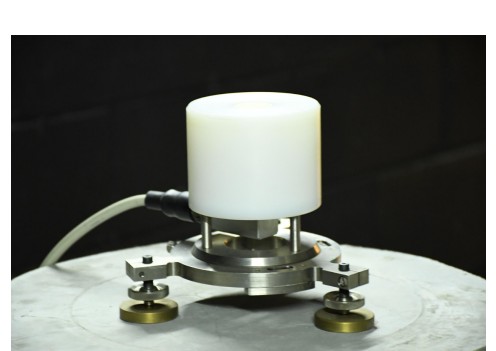 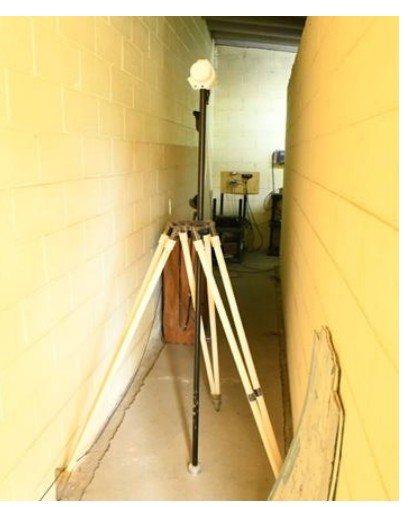 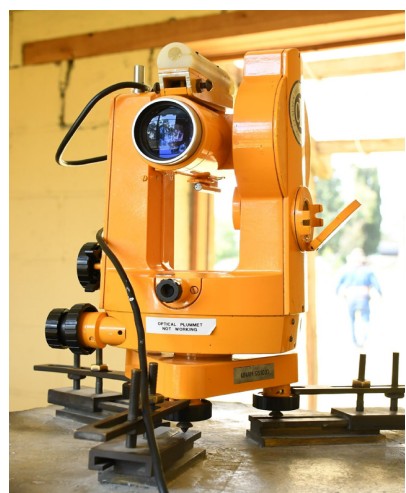

**Figure 8: TEO instrumentation: a) LEMI025 variometer, b) GSM90 magnetometer, c) DI-flux Zeiss Theo 020B**

The operation and distribution of instruments and systems, as well as personnel activities, are described in the operation diagram (Figure 9).

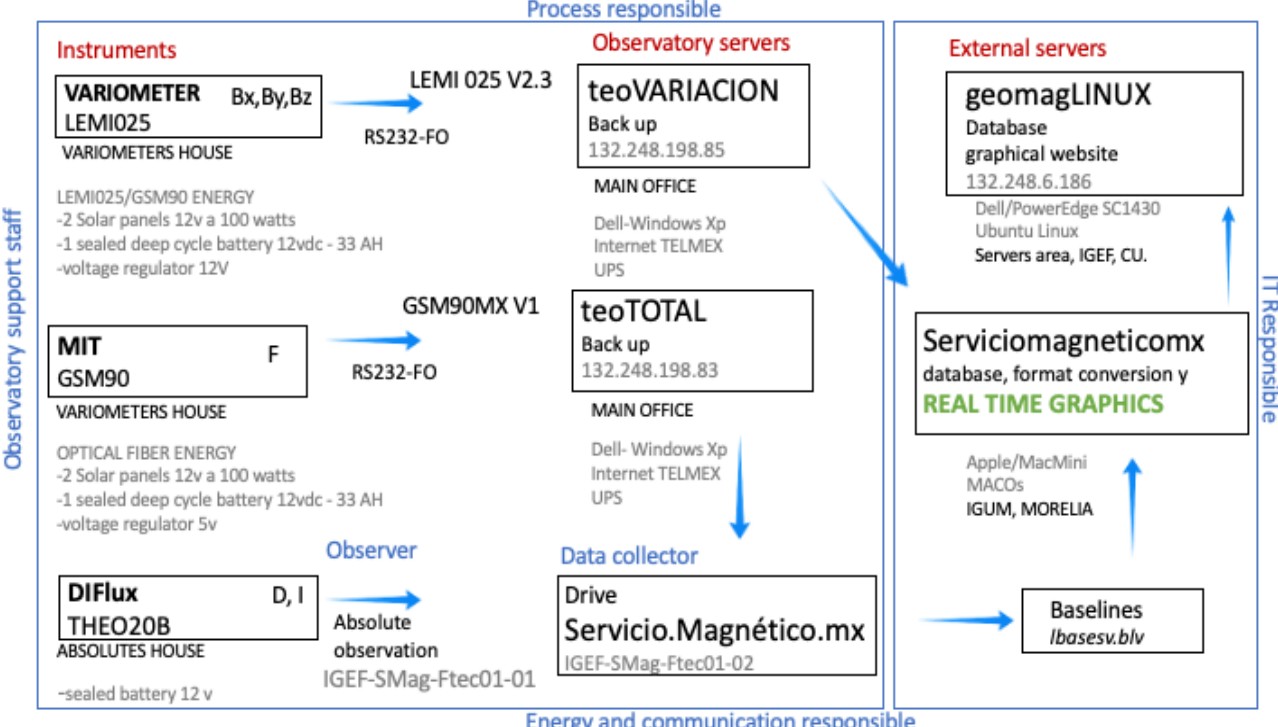

**Figure 9. TEO operation scheme**

The *technical procedure* provides a detailed description of the instruments, generated files, their location, power supply systems, data transmission, and servers.

Variometers

To control the proper operation of the variometer, several continuous tests are performed. The variometer LEMI 025 is in the variometer house in an isolated room; temperature control is performed to check the thermal stability (Figure 10).

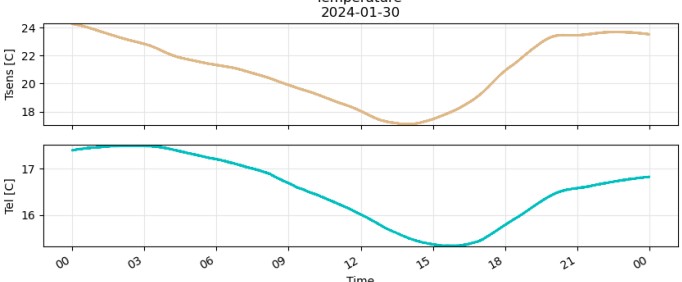

**Figure 10. Temperature at the variometer house. Above temperature at the sensor, below temperature at the electronic unit. (Elaborated with Magpy V2, (Leonhardt et al., 2025))**

It is also compared to a variometer LEMI 417 located outside, 50 m from the variometer house (Figure 11):

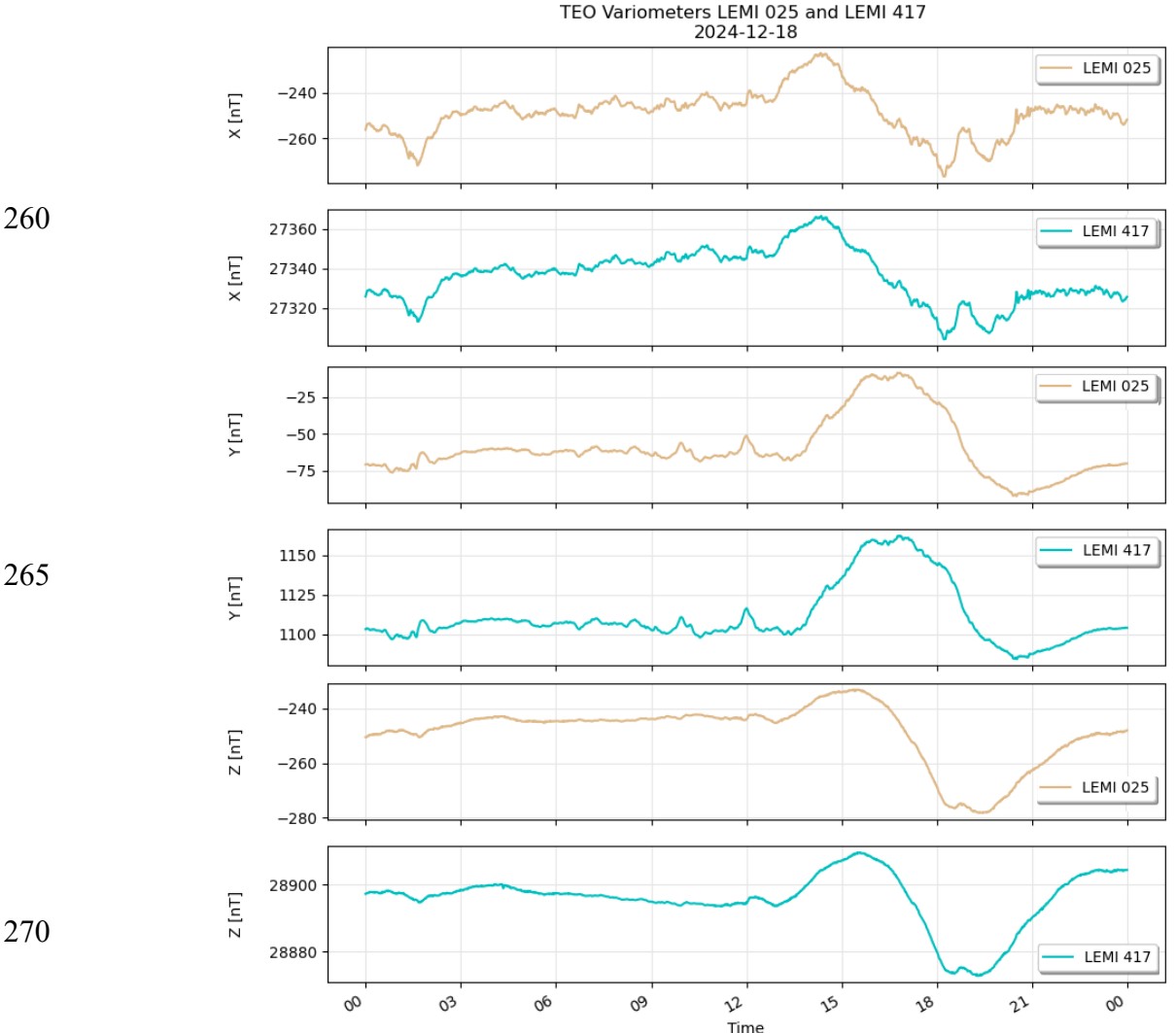

**Figure 11. Variometers comparison: LEMI 025 vs LEMI 417. X, Y, and Z components. (Elaborated with Magpy V2, (Leonhardt et al., 2025))**

It also is compared daily to another observatory at a similar latitude (18.111°N, 293.85°E), San Juan (SJG), and used to observe and remove spikes (Figure 12):

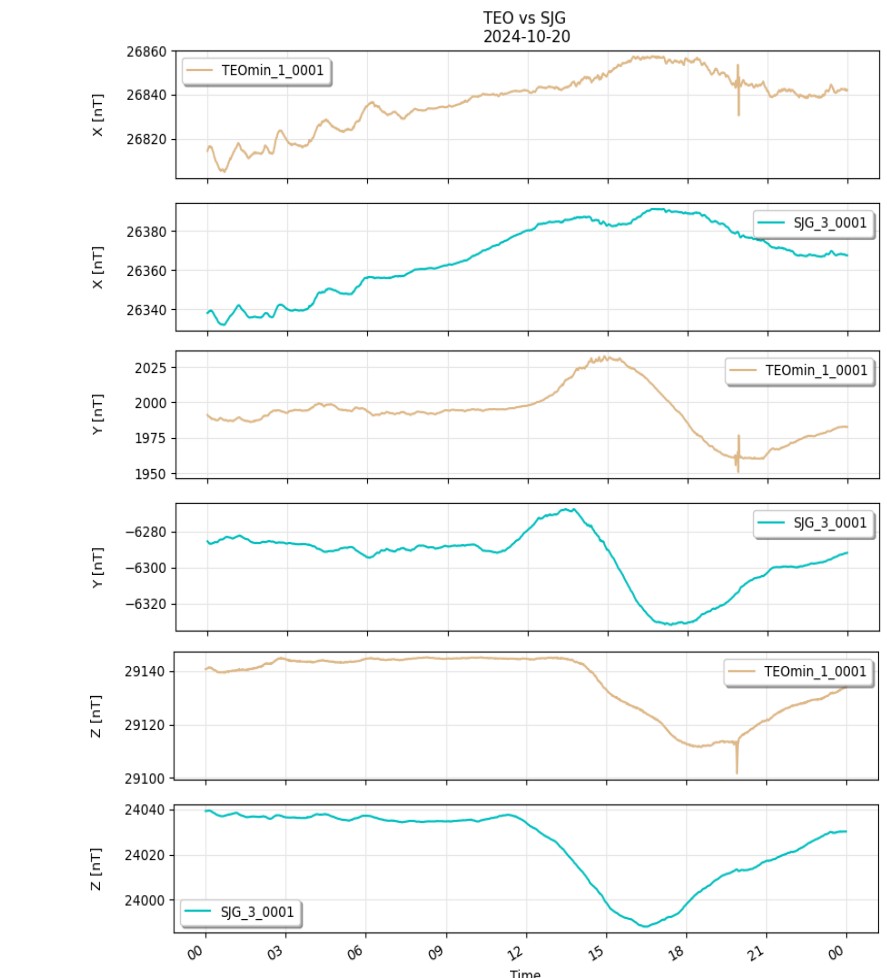

**Figure 12.  Teoloyucan Magnetic Observatory (TEO) compared with San Juan Magnetic Observatory (SJG). X, Y, and Z components. (Elaborated with Magpy V2, (Leonhardt et al., 2025))**

**Absolutes**

It also describes how to perform absolute measurements (Rasson, 2004) and the format (IGEF-SMag-Ftec01-01) where the measurements are recorded (Figure 13). Absolute measurements are performed at least once a week by the observer, avoiding intervals of magnetic activity and to ensure that no magnetic materials, which would cause jumps in baselines are presented, inside or outside the absolute (or variometer) buildings or with the observer (Bracke, 2025).

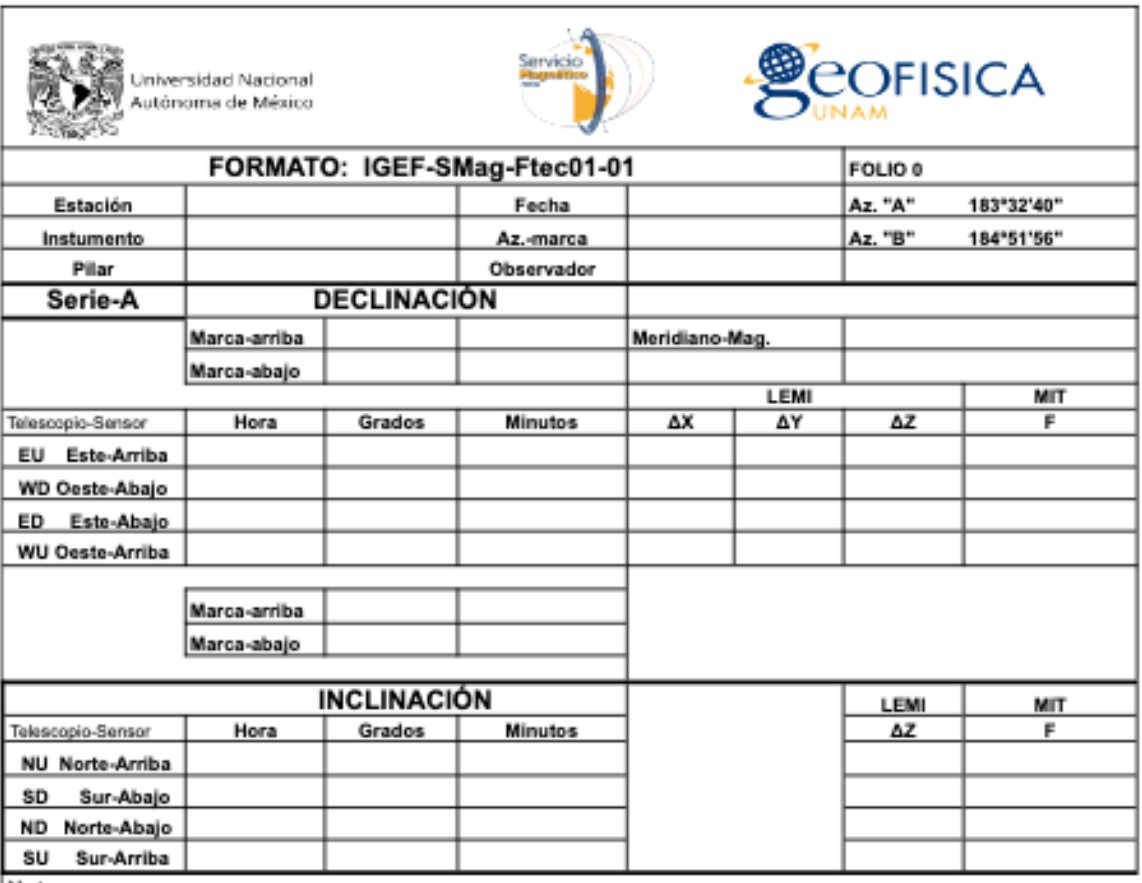

Figure 13. IGEF-SMag-Ftec01-01 form for absolute measurements. Serie A

Then this information is captured by the data collector in the format (IGEF-SMag-Ftec01-02) and double-checked by the process responsible to avoid capture errors or identify spikes. It is also important to consider that a few times a year the absolute measurement is performed with a DI-flux Zeiss Theo 010A from MinGEO to compare results. And that observers participate in the IAGA workshops on Geomagnetic Observatory Instruments, Data Acquisition, and Processing, also to compare the absolute measurements.

With this format the baseline is generated. Complementary, these basevalues are adapted to calculate the observatory baseline using the Magpy software (Leonhardt et al., 2025), along with variometer, and total intensity data (Figure 14):

300

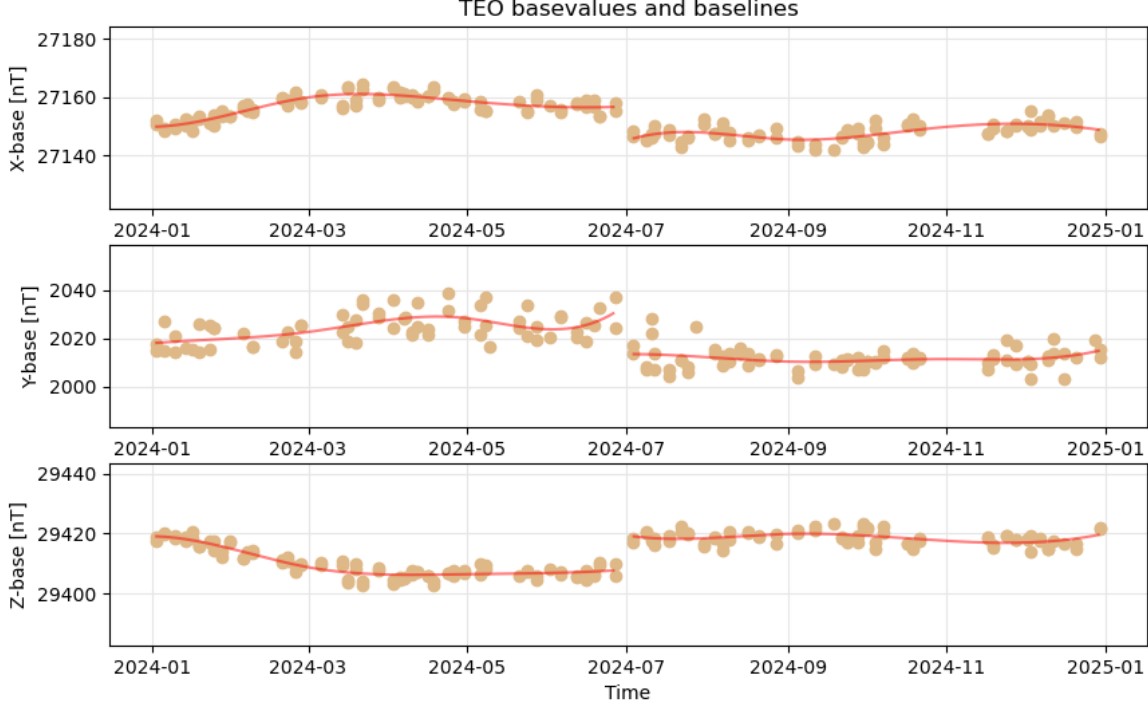

**Figure 14. Basevalues and baselines for TEO 2024. X, Y, and Z components (calculated and elaborated with Magpy V2 (Leonhardt et al., 2025))**

It is important to mention that this procedure includes only the display of geomagnetic data in quasi-real time, i.e., reported data: Vector measurements performed by a continually recording magnetometer are typically variation measurements and variation data, and as considered since Technical Manual V4.0 (1999), variometer data are corrected to near absolute values using the best available baseline estimates at the time of transmission (Bracke, 2025).

All these formats and their location within the Magnetic Service documentation can be found in the technical procedure. Additionally, all the programs and servers that enable data display on the website are also described: those for acquisition, GSM90MX 1.0 (Cifuentes-Nava, 2016) (Figure 15), LEMI V 2.3, and the cron that executes the transmission codes, generation of data in IAGA2002x format, generation of daily, weekly, and 27-day (Carrington rotation) graphs, and their subsequent display on the Magnetic Service website.

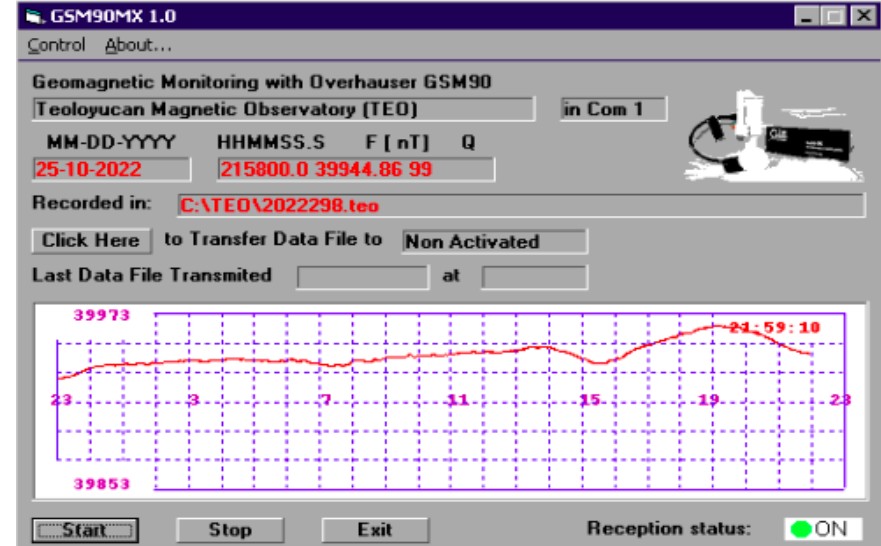

**Figure 15. GSM90MX 1.0 software of the GSM90 magnetometer recording total intensity data (Cifuentes-Nava, 2016).**

On the website, the day before, current day, weekly, and 27-day (Carrington rotation) graphs are shown for the present day, but any previous day can be consulted (Figures 16 and 17).

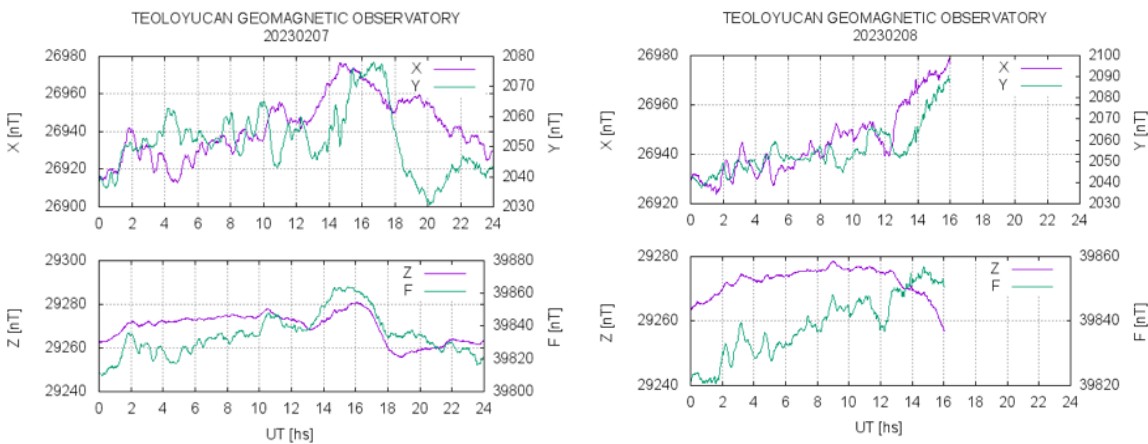

**Figure 16. Plots of the X (purple) and Y (green) components (above), and the Z (purple) and F(green) components (below) displayed on the web page: a) previous day, b) current day.**

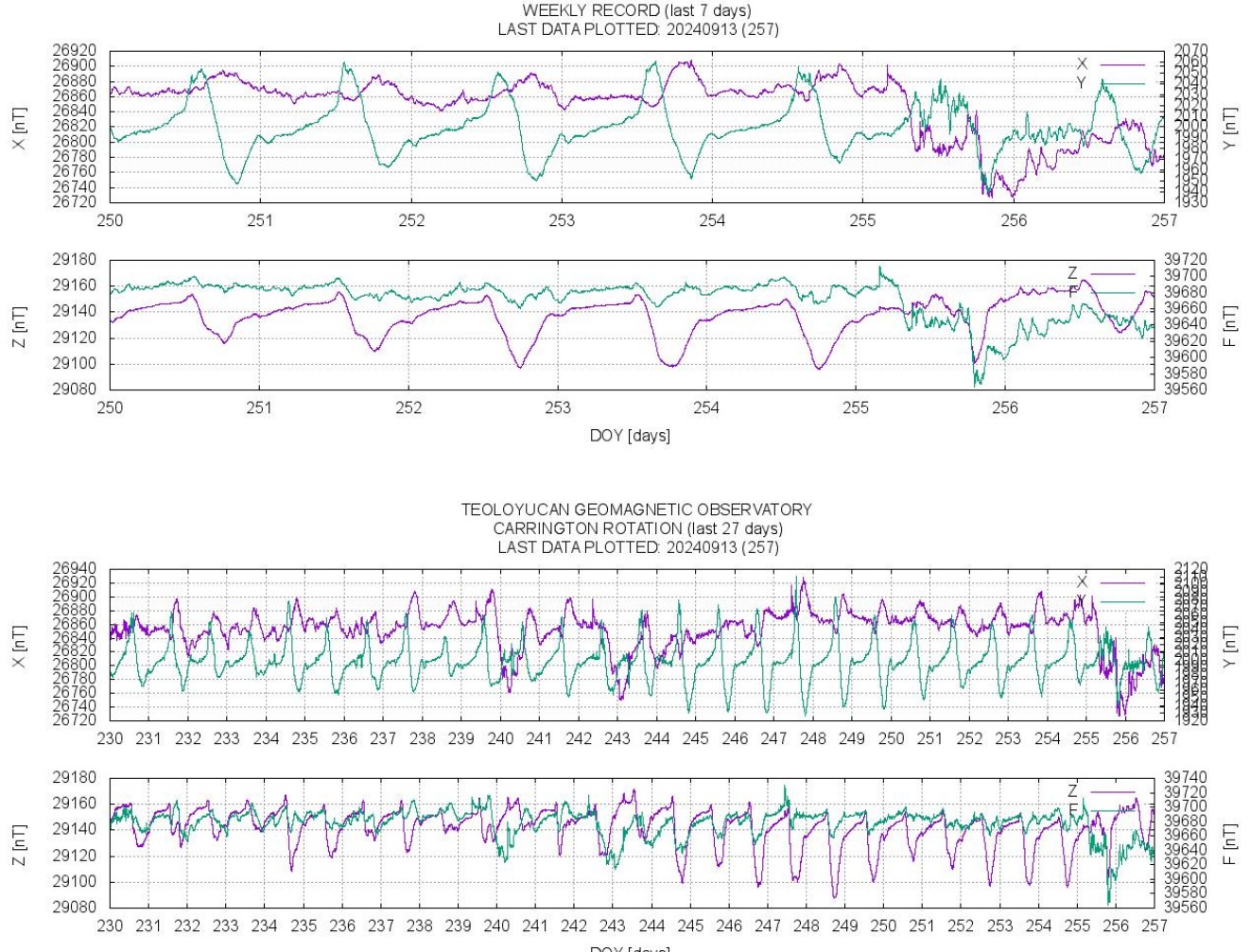

**Figure 17. a) Plot of the last 7 days of the X (purple) and Y (green) components (above), and the Z (purple), and F (green) components (below). b) Plot of the last 27 days (Carrington Rotation) of the X (purple) and Y (green) components (above), and the Z (purple), and F (green) components (below).**

Since the implementation of the QMS, the annual completeness of the data was improved, in accordance with the quality objective, that is, having more than 90%. In 2022 it was 96.49%, 95.66% in 2023, and 99.4 % in 2024.

In the following QR codes (Figure 18), graphical geomagnetic data displayed on the website (Figures 16 and 17) and the ISO 9001:2015 certification document of the geomagnetic data display process in the Magnetic Observatory of Teoloyucan can be consulted.

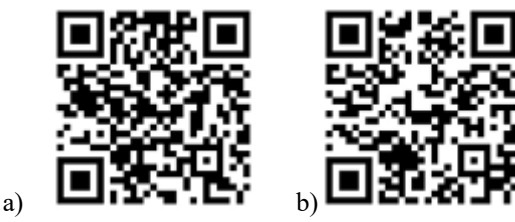

a)  b)

**Figure 18. a) QR code of the graphical geomagnetic data display in the Teoloyucan Geomagnetic Observatory (TEO). b) QR code for the ISO 9001:2015 certification**.

**4. Conclusions**

Certification under the ISO 9001:2015 standard has been designed for industry and business; however, it can be used in academic and research environments, as has been seen in this case. Its usefulness for this case has been mainly to document in detail the operation and generation of a final product under quality standards such as TEO's Graphic Display of Geomagnetic Data (Figure 18a), and for more information about the certificate or the QMS, consult the QR code (Figure 18b).

This certification includes only the display of geomagnetic data in quasi-real time, i.e., reported data: variometer data corrected to near absolute values using the best available baseline estimates at the time of transmission. Based on the results obtained, it is considered to extend the scope of the certification to the procedure for generating adjusted and definitive data. Risks were managed and decreased, contributing to achieve the quality objective of the QMS, which was satisfied with acquiring more than 90% of data per year (96.49% in 2022, 95.66% in 2023, and 99.43% in 2024). Continuous checking of the data and activity log helps to control the data quality.

There are several advantages of implementing a QMS and obtaining an ISO 9001:2015 certification, including setting clear objectives, systematically analyzing risks, creating a culture of continuous improvement, promoting context analysis through a SWOT, and strategic planning based on this knowledge, and semiannual audits. Elaboration of all the documents associated with the operating procedures of a geomagnetic observatory and the continuity of the processes mentioned above guarantee TEO's operation, including both production of data and its availability in real time for users.

**Data Availability**

Data sharing is not applicable to this article, as no data sets were generated or analyzed during the current work.

**Author Contributions**

Conceptualization: Ana Caccavari-Garza; Formal Analysis: Ana Caccavari-Garza, Gerardo Cifuentes-Nava, Armando Carrillo-Vargas, Juan Esteban Hernandez-Quintero; Methodology: Ana Caccavari-Garza, Gerardo Cifuentes-Nava, Armando

Carrillo-Vargas, Juan Esteban Hernandez-Quintero. Project Administration: Adriana Elizabeth González-Cabrera, Ana Caccavari Garza; Software: Gerardo Cifuentes-Nava; Supervision: Ana Caccavari-Garza Validation: Gerardo Cifuentes-Nava. Visualization: Gerardo Cifuentes-Nava. Writing-Original Draft: Ana Caccavari-Garza; Writing-Review & Editing: Ana Caccavari-Garza, Gerardo Cifuentes-Nava, Adriana Elizabeth González-Cabrera, Armando Carrillo-Vargas, Charbeth Lopez-Urías, and Juan Esteban Hernandez-Quintero.

**Competing interests**

The authors declare that they have no conflict of interest.

**Acknowledges**

The authors would like to thank Maricarmen Hernández-Cervantes for her support in the activities associated with the SGC; the Scientific Research Coordination, the Research Quality Management Coordination, and the Institute of Geophysics for
their support in obtaining the ISO 9001-2015 certifications. They also thank the students and operational staff: Jesús Ancira, Nagibe Maroun, and Cristóbal Cifuentes.

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
