# Peer review of "Teoloyucan Geomagnetic Observatory operation over a Quality Management System ISO 9001:2015"

_EGUsphere, 2025_

## Author Comment (AC1)

Dear reviewer,

We would like to thank you for your time and effort in reviewing our paper. We have considered each of your recommendations and we have endeavored to fully address in this resubmitted revision, as we think they contribute to improve this work. Also more specific issues were added to each part of the manuscript, improving some figures (5,6,7), and adding some graph about our data (Figure 10, 11, 12, 14). Along with the basic writing and English, we also believed that there were certain coherence and structural issues that were assessed and solved.

You can find in blue your comments and in black our response.

**General comments:**

The manuscript discusses the implementation of ISO 9001:2015 based on the experiences of the Teoloyucan Geomagnetic Observatory. It analyses the advantages and disadvantages of the standards for industrial quality management systems, especially in the field of regulating operational activities. Although we are informed about improvements in many areas, we do not get a clear answer in two questions: 1. The relationship of the observatory with the INTERMAGNET organization requires clarification, because according to the website https://intermagnet.org/metadata/#/imos the observatory is closed. 2. Regardless of the relationship between the observatory and INTERMAGNET, the question arises as to why the INTERMAGNET Technical Reference Manual (https://tech-man.intermagnet.org/stable/), which provides standards for geomagnetic observatory activities, is not directly mentioned? The quality management system pays special attention to external documents independent of the organization. We do not learn from the manuscript what happened in this specific case. The following sentence is important: "It is important to emphasize that implementing a QMS does not define the operating guidelines themselves, these are given by the international standards already mentioned: the purpose of a QMS is to verify that these guidelines, which are described throughout the QMS, are rigorously followed." In summary: I recommend publishing a list of international documents regulating operations and including them in the references.

After making minor revisions, the manuscript can be published.

1. The relationship of the Observatory with the INTERMAGNET organization is explained.

**Line 61:** The observatory has a continuous record from 1914 until now, it was part of the INTERMAGNET observatories from 2002-2008, but due to instrumental issues, it was not able to maintain data quality, even if it continued operating and sending data to INTERMAGNET until 2021. In 2022 it began operating TEO currently operates with a Ukrainian LEMI025 fluxgate (XYZ) variometer (by Lviv Center of Space Research Institute), a GSM90 Total Intensity (F) magnetometer (by GEM Systems), and a Zeiss Theo 020B DI-flux (Declination (D), Inclination (I)) theodolite with RMIs electronic, following INTERMAGNET standards

(Wienert, 1970; Jankowski and Sucksdorff, Rasson, 2004; Bracke, 2025), and relying on a quality management system to verify the implementation of these standards, with a view to applicate to INTERMAGNET.

2. The Technical Reference Manual it's an important base for the operation of the Observatory, it was cited as its described in the webpage.

   Cite as:

**Line 384:** Bracke, S. (Ed.): INTERMAGNET Operations Committee and Executive Council, INTERMAGNET Technical Reference Manual, Version 5.2.0, 2025.

Along with this, the following reference books and manuals are used:

-Jankowsky, J. and Sucksdorff, C.: Guide for magnetic measurements and observatory practice, IAGA, International Association of Geomagnetism and Aeronomy (IAGA), ISBN: 10-9650686-2-5, 1996.

-Leonhardt, R., Bailey, R., Schovanec, H., Fee, J., Bracke, S., Miklavec, M., and Kompein, N.: MagPy – Analyzing and displaying geomagnetic data (Version 2.0.0). https://doi.org/10.5281/zenodo.15861613, 2025.

-Rasson, J. L.: About Absolute Geomagnetic Measurements in the Observatory and in the Field. Training Booklet for the XIth IAGA Workshop on Geomagnetic Observatory Instruments, Data Acquisition and Processing, Kakioka, Japan, 2004.

Wienert, K. A.: Notes on geomagnetic observatory and survey practice, Earth Sciences, UNESCO, 5, p. 79, 1970.

**Specific comments:**

- Line 40: What considerations were used to select the location of the observatory? The names of several measuring instruments are listed. On what principle did the mentioned devices operate and what quantity did they measure?

Included.

**Line 39:** Teoloyucan site was selected due to the far location to the city, avoiding magnetic noise,

Included.

**Line 51**: It began its operation with Mascart variometers (D, H, Z) that belonged to the previous site. In 1923, the magnetometer Dover 123 and the inclination compasses Fauth 73, Negretti-Zambra 65, and Chasselon 64 were incorporated. Later, in 1931, the set of Eschenhagen photographic recording variometers (D, H, Z) from the Askania House, was acquired and worked until the shift to the digital era (Hernandez Quintero et al., 2018)

- Fig. 2: The figure contains abbreviations that are explained later. Perhaps if any of the abbreviations in the figure were mentioned in the paragraph, the figure could be introduced afterwards.

**Line 69:** Figure 2. TEO's timeline. Location and instrumental changes. Scalar magnetometers: G856(Geometrics), GSM19/90 (GemSystems), POSN129. Variometers: Mascart, Askania-Eschenhagen, LAMA (RMI), FGE(DMI). DI-flux: Ruska, Zeiss Theo 020B.

- Line 55: "In 1993, it was calibrated for the first time with a first-class standard observatory (Friedericksburg, USA). "What does here the calibration mean? Performing a rigorous calibration is unthinkable with such a large distance.

It was referred to an instrumental calibration in the Friedericksburg Observatory, but it was considered better to remove it.

- Fig. 3: The figure contains levels which are explained later. Perhaps the figure could be introduced afterwards.

On the paragraph before Figure 3, was added an explanation of the parts of the Quality Management System to have a better understanding of Figure 3.

**Line 81:** The documentation of the QMS could be summarized in the pyramid shown in Figure 3. Level 1-Quality manual: The mission, goals, objectives, and policy statements. Level 2-Quality system procedure: that describes quality control, validation, and process improvement. Level 3-Instructions (Standard Operation Procedures): operational work instructions and Level 4-Forms: Records, Reports, all the supporting records and forms associated with the operational procedure (Berte and Nevalainen, 1996).

- Fig. 13: The meanings of the colors are not marked and the axis labels are sometimes illegible.

In Figure 13 (now Figure 16 and 17) the meanings of the colors was added:

**Line 337: Figure 17. a) Plot of the last 7 days of X (purple), Y (green) components (above), and Z (purple), and F(green) (below) components. b) Plot of the last 27 days (Carrington Rotation) of the X (purple), Y (green) components (above), and Z (purple), and F (green) (below) components.**

**Technical corrections:**

- Line 48: The article of Berte l. et al. missing from the manuscript.

It is cited in Figure 3 and on the paragraph before this.

**Line 81:** The documentation of the QMS could be summarized in the pyramid shown in Figure 3. Level 1-Quality manual: The mission, goals, objectives, and policy statements. Level 2-Quality system procedure: that

describes quality control, validation, and process improvement. Level 3-Instructions (Standard Operation Procedures): operational work instructions and Level 4-Forms: Records, Reports, all the supporting records and forms associated with the operational procedure (Berte and Nevalainen, 1996).

- Line 330: The article of Kaziliūnas, A. is missing from the manuscrpit.

It was added the missing reference.

**Line 204:** Besides being a standard requirement, audits enable confirming if the QMS complies with both its own and the standard's requirements, as well as whether its deployment and operation are successful. Even though it could seem like a laborious process at first, having these systematic evaluations allows one to ensure that every item outlined in the technical procedure and the QMS is promptly addressed (Kaziliūnas, 2010),

- Line 35: Rasson et al, 2011 has an erroneous date (2001) in the references.

It was corrected:

**Line 407:** Rasson, J. L., Toh, H., and Yang, D.: The Global Geomagnetic Observatory Network, in: Geomagnetic Observations and Models, edited by Mandea, M. and Korte, M, IAGA Special Sopron Book Series 5Springer Netherlands, 1-25, https://doi.org/10.1007/978-90-481-9858-01, 2011.

- Line 60: The meaning of UNAM is not clear to me.

The meaning of UNAM was included.

**Line 38:** In Mexico, the only existing geomagnetic observatory is managed by the Magnetic Service of the Geophysics Institute of the National Autonomous University of Mexico (UNAM).

- Line 65: The meaning of QMS is not defined.

The meaning was added.

**Line 12:** Given the nature of Quality Management Systems (QMS) based on ISO 9001:2015, we consider that their implementation in a geomagnetic observatory, can be a valuable tool that allows monitoring the follow-up of international standards and ensuring their proper operation, thus guaranteeing high-quality geomagnetic data.

- Line 115: SWOT is absolutely not defined in this paragraph.

The meaning of SWOT was defined.

**Line 77:** continuous review of *strengths, weaknesses, opportunities* and *threats (SWOT)*

---

## Author Comment (AC2)

Dear reviewer,

Based on your general comments, we would like to express our gratitude for recognizing that this effort can complement INTERMAGNET standards and benefit the global community working on long-term geomagnetic observations. We have considered each of your recommendations and made an effort to address them since we think that supporting the article's statements about data quality enhances this work. Along with the basic writing and English, we also believed that there were certain coherence and structural issues that were assessed and solved.

You can find in blue your comments and in black our response.

According to your comments those issues were added:

GENERAL COMMENTS

However, some aspects of the manuscript need improvement. The discussion of benefits is often too general and theoretical. Although many advantages of the QMS are listed (e.g., clear goals, risk analysis, continuous improvement), the paper lacks concrete, measurable examples that show how these benefits translated into better data quality, more efficient operations, or problem-solving at Teoloyucan. It would be necessary to include specific evidence - such as improvements in baselines determination.

Recommendation: Accept with major revisions, focused on:

- Adding specific, measurable outcomes to support the claimed benefits,
- Correcting structural inconsistencies (especially regarding figures and references),
- Improving the overall clarity and language quality of the text.

Several specific aspects were included, related with both the operation and the data.

-Figure 5 was changed to show in detail the possible risks during the observatory operation and how these risks can be managed to reduce the consequences:

| Risk | Event that can cause | Probability/frequency | Consequence/damage | Risk assessment (F vs D) | Mitigation actions | Occured | Consequence | Risk reassessment (F vs D) | New mitigation actions |
|---|---|---|---|---|---|---|---|---|---|
| Pandemic/strikes | No access to the facilities | Probable | Critical | 9 | Alternate center for downloading and processing information | No | None | 6 | None |
| Lack of supplier inventory | Not being able to replace faulty equipment/Stop recording data | Probable | Critical | 9 | Look for another provider | No | None | 6 | None |
| Long acquisition times | Not being able to replace faulty equipment/Stop recording data | Probable | Critical | 9 | Constant communication with the administration | No | None | 6 | None |
| Extreme phenomena | Stop recording data | Remote | Catastrophic | 8 | Backup equipment | No | None | 8 | None |
| Theft/vandalism | Stop recording data | Remote | Catastrophic | 8 | Surveillance and access control | No | None | 8 | None |
| Magnetic noise | Noisy data | Remote | Critical | 6 | Constant data quality control and surveys to detect Magnetic objects | No | None | 6 | None |
| Instrumental damage | Stop recording data | Probable | Catastrophic | 12 | Backup instruments | No | None | 8 | None |
| Lack of internet for streaming | Real-time reporting of data cannot be performed | Probable | Critical | 9 | Have a second way to download data | Yes | It affected the website display time | 9 | Have a second way to download data and request support from RedUNAM |
| Logical errors on the server | | Probable | Critical | 9 | Second server that can perform the processing | No | None | 9 | None |
| Power outage | | Probable | Critical | 9 | Backup power system in case of prolonged power outages | No | None | 9 | None |

**Figure 5. Reduced risk matrix of the Magnetic Service**

-Figure 6 was improved to detail the activities and responsibilities of the staff:

[Figure]

**Figure 6. Magnetic Service organizational chart**

-Instrumental control and logbook were described with more specific activities. If considered additional charts could be added to should the flagging of the activities and events on the observatory graphs:

**Line 164 -169**: Within the *management documents*, in the part corresponding to *instrument control,* there is a format (SMag IGEF-FPG02-01) that contains the inventory of instruments and servers, their characteristics, and their location. There is also a format (SMag IGEF-FPG02-03) that tracks instrument maintenance, such as routine cleaning of servers, antivirus revision, delta F measurement, theodolite maintenance, comparison with Theo 010A DI-flux, cleaning, verification and adjustment, and internal cleaning of LEMI 025 memory, among others (Bracke, 2025), The instruments maintenance format, includes the equipment, activity, responsible, date of last maintenance and date for the next, it is also link to a calendar to remind the activity.

Complementary to the records of the equipment and its maintenance, there is a logbook to record failures that occur in the equipment or systems, record any maintenance activity, either preventive or corrective, and record changes or events that produce disturbances in data that affect data quality.

-Figure 7 was improved:

[Figure]

**Figure 7. a) TEO's schematic drawing and b) aerial photo**

-A variometer section was added to describe the specific activities of quality control: thermal stability (Figure 10), comparison with another variometer (LEMI417) in the observatory (Figure 11) and with other close latitude observatory, San Juan (SJG)(Figure 12).:

**Line 246**: To control the proper operation of the variometer, several continuous tests are performed. The variometer LEMI 025 is in the variometer house in an isolated room, temperature control is performed to check the thermal stability (Figure 10).

[Figure]

**Figure 10. Temperature at the variometer house. Above temperature at the sensor, below temperature at the electronic unit. (Elaborated with Magpy V2, (Leonhardt et al., 2025))**

It is also compared to a variometer LEMI 417 located outside, 50 m from the variometer house (Figure 11):

[Figure]

**Figure 11. Variometers comparison: LEMI 025 vs LEMI 417. X, Y, Z components. (Elaborated with Magpy V2, (Leonhardt, et al., 2025))**

It also is compared daily to another observatory at a similar latitude (18.111°N, 293.85°E), San Juan (SJG) and used to observe and remove spikes (Figure 12):

[Figure]

**Figure 12.  Teoloyucan Magnetic Observatory (TEO) compare with San Juan Magnetic Observatory (SJG).  X, Y, Z components. (Elaborated with Magpy V2, (Leonhardt et al., 2025))**

-Also, was added a paragraph about absolute measurements, inter comparison with other DI-flux measurements (Theo010A) and with other observatories instruments in the IAGA Workshop. And a description and graph (Figure 14) of basevalues and baselines calculation utilizing the Magpy V2 software.

**Line 29:** It also describes how to perform absolute measurements (Rasson, 2004) and the format (IGEF-SMag-Ftec01-01) where the measurements are recorded (Figure 13). Absolute measurements are performed at least once a week by the observer, avoiding intervals of magnetic activity and to ensure that no magnetic materials, which would cause jumps in baselines are present, inside or outside the absolute (or variometer) buildings, or with the observer (Bracke, 2025).

**Line 297:** Then this information is captured by the data collector in the format (IGEF-SMag-Ftec01-02) and double-checked by the process responsible to avoid capture errors or identify spikes. It is also important to consider that few times a year the absolute measurement is performed with a DI-flux Zeiss Theo 010A from MinGEO to compare results. And that observers participate in the IAGA workshops on Geomagnetic Observatory Instruments, Data Acquisition and Processing also to compare the absolute measurements.

With this format the baseline is generated. Complementary, these basevalues are adapted to calculate the observatory baseline using the Magpy software (Leonhardt et al., 2025), along with variometer, and total intensity data (Figure 14):

[Figure]

**Figure 14. Basevalues and baselines for TEO 2024. XYZ (calculated and elaborated with Magpy V2 (Leonhardt et al., 2025))**

It is important to mention that this procedure includes only the display of geomagnetic data in quasi-real time, i.e. reported data: Vector measurements performed by a continually recording magnetometer are typically variation measurements and variation data and as considered since Technical Manual V4.0 (1999), variometer data are corrected to near absolute values using the best available baseline estimates at the time of transmission (Bracke, 2025)

- Conclusions were modified specifying benefits in the operation and quality data. Including the percentage of annual data (96.49%-2022, 95.66%-2023, 99.43%-2024).

**Line 348:  4. Conclusions**

Certification under the ISO 9001:2015 standard has been designed for industry and business; however, it can be used in academic and research environments, as has been seen in this case. Its usefulness for this case has been mainly to document in detail the operation and generation of a final product under quality standards such as TEO's Graphic Display of Geomagnetic Data (Figure 18a), and for more information about the certificate or the QMS, consult the QR code (Figure 18b).

This certification includes only the display of geomagnetic data in quasi-real time, i.e. reported data: variometer data corrected to near absolute values using the best available baseline estimates at the time of transmission. Based on the results obtained, it is considered to extend the scope of the certification to the procedure for generating adjusted and definitive data. Risks were managed and decreased, contributing to achieve the quality objective of the QMS, was satisfied acquiring more than 90% of data per year (96.49%-2022, 95.66%-2023, 99.43%-2024). Continuous checking of the data and activity log, helps to control the data quality.

There are several advantages of implementing a QMS and obtaining an ISO 9001:2015 certification including setting clear objectives, systematically analyzing risks, creating a culture of continuous improvement, promoting context analysis through a SWOT, and strategic planning based on this knowledge, and semiannual audits.  Elaboration of all the documents associated with the operating procedures of a geomagnetic observatory and the continuity of the processes mentioned above guarantee TEO's operation, including both production of data and its availability in real time for users.

- Additionally, the structure of the manuscript is somewhat inconsistent. Several figures (1, 2, 4, 6, 12, and 13) are not referenced in the text, and some citations are unclear or incomplete. These issues have to be addressed to improve the clarity and coherence of the article.

References to Figure 1, 2,4,6, 12 and 13 (now 16 and 17) were added to text:

**Line 37**: It has operated in the town of Teoloyucan, State of Mexico (Figure 1), since August 1914,

**Line 49**: The present operational buildings were constructed during this period (Figure 2).

**Line 136:** As show in Figure 4, this analysis offers different possibilities;

**Line 156:** For the operation of the Magnetic Observatory of Teoloyucan, the Magnetic Service defined the following organization chart (Figure 6):

**Line 325:** On the website, the day before, current day, weekly, and 27-day (Carrington rotation) graphs are shown for the present day, but any previous day can be consulted (Figures 16 and 17).

SPECIFIC COMMENTS

- Line 41. Reference to the wrong figure, because Figure 3 is about "Structure of a Quality Management ..."

Corrected, it was Figure 2 and the place of the reference was changed.

**Line 49:** The present operational buildings were constructed during this period (Figure 2).

- Line 53. What does the abbreviation DFI mean? It is not explained anywhere.

Explained and the beginning of the Introduction:

**Line 30:** The components of the magnetic field vector, include total intensity (F), horizontal (H), vertical (Z), north (X), east (Y), declination (D), and inclination (I).

- Line 64. The abbreviation DI is introduced but not used later.

A paragraph was added to name the components of the magnetic field vector.

**Line 30:** The components of the magnetic field vector, include total intensity (F), horizontal (H), vertical (Z), north (X), east (Y), declination (D), and inclination (I).

- Line 65. This is the first time the abbreviation QMS is used. It's better to write the full name here with "QMS" in parentheses, even though the full name appears in the title of the article.

Corrected. Quality Management System (QMS)

**Line 12**: Given the nature of Quality Management Systems (QMS) based on ISO 9001:2015

- Lines 79,84. This item is missing from the References.

The reference was added:

**Line 390**: Godinez-Mendez, W.: The Implementation of Quality Management Systems in Law Libraries: Esquemas de trabajo híbrido y nuevos escenarios internacionales en las bibliotecas jurídicas, edited by Hernández Pacheco, Federico. ISBN: 978-607-30-7824-5, 2023.

- Line 221.There should probably be a reference to Fig. 7 here.

Reference to Figure 7 corrected.

**Line 225**: A schematic drawing of the Teoloyucan Magnetic Observatory is shown in Figure 7: absolute and variometer houses, two auxiliary huts, and an office.

- Line 225. There is an unnecessary comma after the word "hut." The whole sentence is complicated and hard to read.

Corrected. The paragraph was rewritten more clearly, and Figure 7 has been updated with the right names.

**Line 228**: The observatory was built following the established standards: the absolute observations house has two pillars for measurements, is built of non-magnetic materials, and has a window to observe the azimuth mark, located 70 m away; the variometer house consists of an outer corridor and an inner isolated room. The inner room is equipped with five solid pillars, is built of non-magnetic materials. In one of those pillars is the LEMI 025 sensor. In the outer corridor is located the electronics of the LEMI 025, in the norther part the GSM90 magnetometer, and in the other side the power and transmission systems. The instruments are powered by solar panels (Figure 7). The data are transmitted by optic fiber to the office since 2022, to avoid problems due to currents induced by lightning strikes.  Instruments are shown in Figure 8.

Line 314   It's worth making the whole References section consistent with the chosen style, as much as possible.

All References were checked and corrected.

- Line 315. Punctuation. It's worth to add the DOI: https://doi.org/10.1093/labmed/27.6.375

Added

**Line 382**: Berte, L., and Nevalainen, D.: Managers' Documentation Pyramid for a Quality System. V. 27, N. 6 Laboratory Medicine, https://doi.org/10.1093/labmed/27.6.375, 1996.

- Line 331.This item is listed in the References but is not cited anywhere in the article.

It was added The missing reference was added:

Line 202 Besides being a standard requirement, audits enable confirming if the QMS complies with both its own and the standard's requirements, as well as whether its deployment and operation are successful. Even though it could seem like a laborious process at first, having these systematic evaluations allows one to ensure that every item outlined in the technical procedure and the QMS is promptly addressed (Kaziliūnas, 2010),

Line 338          Should be probably: Rasson, J.L., Toh, H., Yang, D. (2011). The Global Geomagnetic Observatory Network. In: Mandea, M., Korte, M. (eds) Geomagnetic Observations and Models. IAGA Special Sopron Book Series, Vol 5. Springer, Dordrecht. https://doi.org/10.1007/978-90-481-9858-0_1

Corrected.

Line 409 Rasson, J. L., Toh, H., and Yang, D.: The Global Geomagnetic Observatory Network, in: Geomagnetic Observations and Models, edited by Mandea, M. and Korte, M, IAGA Special Sopron Book Series 5Springer Netherlands, 1-25, https://doi.org/10.1007/978-90-481-9858-01,  2011.

TECHNICAL CORRECTIONS

- Lines 30,65,85   The author "Brake" is missing in the References. It is probably meant to be "Bracke."

    Corrected

 Line 30, 66, 95, 165, 294  (Bracke, 2025)

- Line 35              There is a missing dot after "al"

Corrected:

Line 35 Rasson et al., 2011

- Line 81              It should be "Figure 3" (the dot is not needed)

Corrected:

Line 90 Figure 3:

- Line 283              There is a missing comma after the word "however"

Added

Line 349 however, it can be used in academic and research environments,

Line 339          There is missing "d" in "edited"

Added

Line 410 edited by Mandea, M. and Korte,